# Assessing the role of biomarker feedback in a 12-week community weight management programme among overweight men: A pilot study

Daniel Grant[1], Joshua Smith[1], Lindsay Bottoms[2]*

1 Medichecks, Ranch House, Chapel Lane, Bingham, Nottingham, United Kingdom, 2 Centre for Research in Psychology and Sport Sciences, University of Hertfordshire, Hatfield, United Kingdom

* l.bottoms@herts.ac.uk

**Data Availability Statement:** The data are available at https://doi.org/10.18745/ds.27470.

## Abstract

### Background

The primary objective of this study was to examine the feasibility of recruitment, retention, and delivery of biomarker feedback among men partaking in Shape Up, a physical activity for health programme. Secondarily, it explored the potential effects of biomarker feedback on participants' adherence and motivation levels.

### Methods

In this two-arm non-randomised pilot study, 46 men (mean±SD age 46.0±8.6 years) participating in the 12-week Shape Up programme were assigned to a control group or biomarker feedback group. Biomarker feedback consisted of 3 blood test panels: at baseline, 12 weeks, and 24 weeks (follow-up), each with a doctor's report. Both groups completed questionnaires (Short Active Lives Survey [SALS], Treatment Self-Regulation Questionnaire [TSRQ], and EQ-5D-5L) to gauge levels of motivation and engagement. The mean difference was calculated between baseline and programme end. Recruitment, retention, and attendance rates were determined.

### Results

Mean recruitment (approach-to-consent), retention, and attendance rates were 96.3%, 92.3%, and 83.7% in the control group, and 85.2%, 86.2%, and 81.5% in the biomarker feedback group, respectively. Of biomarker feedback group participants, 86.4% attended their 12-week blood test and 45.5% attended their follow-up blood test. The mean 12-week differences (95% CI) for the control group and biomarker feedback group were 138.1 (2.7, 273.5) and 467.3 (249.4, 685.2) for active minutes per week, 0.2 (-0.8, 1.2) and 0.4 (0.0, 0.8) for autonomous TSRQ domains, 0.2 (-0.3, 0.8) and 0.8 (0.1, 1.4) for controlled TSRQ domains, and 19 (12.7, 26.5) and 27.2 (19.8, 34.6) for EQ-5D-5L scores, respectively.

**Funding:** Medichecks supported this study via salary for JS and DG. The specific roles of these authors are articulated in the 'author contributions' section. Medichecks also provided the blood tests and doctors' reports used in the study free of charge. The funders had no role in study design, data collection, and analysis, decision to publish, or preparation of the manuscript.

**Competing interests:** The authors have read the journal's policy and have the following competing interests: JS and DG are paid employees of Medichecks, LTD. This does not alter our adherence to PLOS ONE policies on sharing data and materials.

**Abbreviations:** BMI, Body mass index; CI, Confidence interval; EDTA, Ethylenediaminetetraacetic acid; EQ-5D-5L, EuroQol-5D-5L; FC CSE, Football Club Community Sports and Education; GP, General practitioner; HbA1c, Haemoglobin A1c; HDL, High-density lipoprotein; Hs-CRP, High-sensitivity C-reactive protein; IQR, Interquartile range; ITT, intention-to-treat; LDL, Low-density lipoprotein; MD, Mean difference; ND, No data; NHS, National Health Service; RCT, Randomised controlled trial; SALS, Short Active Lives Survey; SD, Standard deviation; SP, Social prescribing; SPIRIT, Standard Protocol Items: Recommendations for Interventional Trials; TSRQ, Treatment Self-Regulation Questionnaire; UKAS, United Kingdom Accreditation Service.

## Conclusion

Biomarker feedback was well-received by participants and deemed feasible, with high recruitment and retention rates for the duration of the 12-week programme. Biomarker feedback may affect aspects of motivation but did not appear to influence adherence to the programme. Biomarker data may provide additional evidence of programme efficacy. Important design considerations are provided for definitive larger-scale trials.

## Introduction

Over a quarter of the population in the United Kingdom is obese [1]. Obesity increases the risk of other health problems, reduces life expectancy, and impacts people's quality of life [2]. It remains one of the nation's greatest long-term health challenges and the National Health Service (NHS) Long Term Plan recognises a need to radically reconsider how health services are delivered [3, 4].

One of the government's strategies is to focus on developing community-based initiatives, such as tier 2 weight management services, to provide targeted support to patients at greatest risk [4]. These services typically involve a multidisciplinary team and personalised interventions to address both physical and psychological aspects of weight management. Research demonstrates these initiatives are effective in supporting weight loss at a population level and generating lasting health improvements, however, their success is largely dependent on recruiting and maintaining participants in the programme [5, 6]. Public health data from 2021 to 2022 showed that tier 2 weight management programmes had enrolment and completion rates of 65% and 35%, respectively, with 15% of patients losing more than 5% of their initial body weight [7].

Similar recruitment figures are seen for exercise referral schemes, another form of community-centred intervention whose primary focus is to increase activity levels through programmes tailored to the patient's needs. Two systematic reviews revealed that the uptake and completion rates of UK exercise referral schemes may be as low as 23–66% and 12–42%, respectively, the consequences of which may include suboptimal outcomes, wasted resources, and greater health inequalities [8–10]. The absolute risk reduction of these schemes was found to be small (17 referrals for 1 to become moderately active), at least partly due to poor enrolment and adherence rates, highlighting the need to prioritise improvements in these areas to enhance programme effectiveness [9, 11].

Factors affecting enrolment and adherence in community programmes have been partially explored. These include age and gender of participants, patients' trust in GPs, programme length, degree of flexibility and choice, the skill level of the activity leader, and the perceived benefit of the intervention [12, 13]. However, it is important identifying other ways to further refine these programmes and optimise investment [12].

The use of biomarkers may help to tackle some of these challenges. Biomarkers are measured characteristics, such as haemoglobin A1c (HbA1c), used to assess an individual's health status and provide evidence of changes in physiology [14]. When results are shared with patients as actionable advice (biomarker feedback), it may benefit weight loss programmes in several ways. Firstly, well-informed patients with access to health information are generally more engaged in the management of their own health [15, 16]. Recent evidence shows that high-retention programmes tend to feature a greater number of behaviour change techniques, including self-monitoring, positive reinforcement, and biofeedback [17]. Consequently,

biofeedback has been established as a key recommendation in weight management programmes to promote participant retention [11]. While many forms of biofeedback, such as blood pressure monitoring, are already established in programmes, few have explored the potential impact of blood biomarkers. Mitchell et al. specifically investigated the effects of both limited and enhanced biomarker feedback on physical activity, weight loss, and blood pressure as part of a public health initiative [18]. Only small improvements were noted in blood pressure in the enhanced biomarker feedback group, but these participants received minimal counselling with their results, the biomarkers selected were minimally responsive to lifestyle changes, and there was no control group for comparison. Therefore, further evidence is needed to clarify any potential effect of biomarker feedback within these contexts. Secondly, unlike some healthcare measures, biomarkers are objectively quantifiable, less prone to bias, and may detect physiological changes beyond an individual's perception, adding to the overall assessment of physical health [19, 20]. If biomarker data can provide further efficacy of a programme, participants may be more likely to join or engage, since patients are more likely to participate in a programme if they believe it will be beneficial [21, 22]. Thirdly, biomarker data may provide a basis for tailoring preventative interventions to improve their cost-effectiveness and further benefit the patient [23, 24]. For example, raised levels of predictive biomarkers, like HbA1c, may highlight patients who would benefit from additional interventions suited to their needs [14]. Tailoring programmes to the individual has been shown to support long-term adherence [25, 26].

There is also a need to clarify effectiveness of programmes to strengthen health economic evidence [27]. This is particularly important for social prescribing (SP) programmes [28]. SP is a non-medical referral option for healthcare workers that aims to link patients to a range of local services, including physical activity programmes [29]. It forms a part of delivering personalised care, empowering patients to choose options that meet their physical, social, and emotional needs [30, 31]. NHS bodies have recognised the positive impact that SP may play in community-based support and aim to accommodate 900,000 referrals by 2024 [4, 31]. Unfortunately, there are few high-quality systematic reviews that investigate the effectiveness (and cost-effectiveness) of these programmes [32, 33]. Much of the evidence stems from interventional trials using unstandardised, self-reported, or qualitative outcomes [34, 35]. While important, these findings make it difficult to quantify success in a consistent way and estimate cost-effectiveness of the intervention prior to wider-scale adoption [32, 35]. Biomarker data may help to strengthen objective evidence of a programme's success, whilst also predicting future health service utilisation and associated costs [23].

In summary, we hypothesise that personalised and detailed biomarker feedback may enhance motivation and adherence within community physical activity programmes, as well as strengthening data to measure efficacy and cost-effectiveness of such interventions.

## Study aims and objectives

This pilot study aims to investigate the feasibility and acceptability of integrating blood testing into a community-based weight management programme for overweight men. The results provide data to estimate the parameters required to design a definitive randomised controlled trial (RCT).

The primary objectives of the study were to:

1. Assess the feasibility of recruiting and retaining participants in the trial

2. Evaluate the feasibility of collecting valid outcome measures

3. Explore the acceptability of the intervention for participants

4. Estimate the sample size required to adequately power a future RCT

Secondary outcome measures were devised to ensure the impact of biomarker feedback can be assessed appropriately in future RCTs.

The secondary objectives of the study were to:

1. Explore the potential effect of biomarker feedback on clinical outcomes and behaviour change within weight management programmes

2. Explore the potential role of biomarker data in strengthening evaluations of weight management programme efficacy and cost-effectiveness

## Methods

### Trial design and setting

The 6-month parallel two-arm non-randomised pilot study investigated the feasibility of integrating biomarker feedback into a community weight management course. Watford Football Club Community Sports and Education (FC CSE) Trust's Shape Up programme (https://www.watfordfccsetrust.com/project/shape-up/) is a community initiative based in Hertfordshire and Harrow, UK, designed to help men lose weight and learn more about healthy lifestyle behaviours. The initiative is for men aged 18 to 65 years old with a body mass index (BMI) of 30 or over (or a BMI of 28 or over for men who are black or from ethnic minority backgrounds). Men can either self-refer to the programme or sign up through their general practice. Participants were recruited opportunistically from two sites, Westfield and Borehamwood, which formed the intervention and control groups, respectively.

The study was reported in accordance with relevant guidelines, including the Transparent Reporting of Evaluations with Nonrandomized Designs (TREND) checklist [36] (S1 Appendix) and relevant items from the Standard Protocol Items: Recommendations for Interventional Trials (SPIRIT) checklist [37].

### Participants

**Recruitment.** The recruitment period ran from the 18th to the 31st January 2019 and ceased due to study timelines. The follow-up period ran from the 22nd to the 26th July 2019. All recent joiners of the Shape Up programme at the Westfield and Borehamwood sites were given an oral presentation detailing the purpose and outline of the study, delivered by one of the research staff. This included details of the intervention and which site this would concern. Participants were also issued a participant information booklet. Interested participants registered their interest with the researcher in person.

**Eligibility criteria.** Interested participants were issued a pre-screening questionnaire to obtain baseline characteristics (age, ethnicity, weight) and to identify any contraindications to exercise or the study intervention. Results were collected and reviewed by research staff to determine the final study groups according to the following inclusion and exclusion criteria.

Inclusion criteria:

1. Male aged 18–65

2. BMI $\geq$30 (or BMI $\geq$28 if from black or ethnic minority background)

3. Physical ability to take part in the physical component of the programme

4. Willing and able to travel to the site for the intervention

5. Capacity to give informed consent

Exclusion criteria:

1. Absolute physical contraindications to exercise according to the exercise preparticipation health screening logic model by Riebe et al. [38]

2. Inability to provide informed consent due to acute mental illness, severe cognitive impairment, or severe intellectual disability

3. Insufficient fluency in English to provide informed consent or complete exit interviews

4. Significant needle phobia

After fulfilling the above criteria, each site was randomly appointed as either the control or intervention group.

**Sample size.**   As this was a feasibility study, a formal sample size was not calculated. All recent joiners of the Shape Up programme at Borehamwood and Westfield sites were approached to recruit as many participants as possible into the study.

**Consent, ethical approval, and clinical trial registration.**   Written informed consent was obtained from all participants after details of the trial were explained. Ethical approval was granted by the University of Hertfordshire with approval ID LMS/SF/UH/03620. The authors confirm that all ongoing and related trials for this intervention are registered (clinical registration number: ID NCT03809871).

## Control and intervention groups

**Control group.**   Watford FC CSE's Shape Up programme is a free 12-week weight management programme for overweight men, accessible through their GP or via direct contact. Participants attended a weekly 90-minute session, comprising of an educational component and a physical activity component. Initially, the education component was longer, and as the programme progressed, this reduced until the last session was entirely physical. The educational component was led by a healthy lifestyle project officer (Level 4 Register of Exercise Professionals) and included topics such as healthy eating and the effects of physical activity and stress on wellbeing. The exercise programme was performed at the fitness facility and supervised by an exercise instructor. It included whole-body activities (circuit training) and group exercises (e.g. football and other team games). Participants were weighed at the beginning (baseline) and end of the programme (week 12), and at a follow-up session (week 24).

All participants were asked to complete three questionnaires at baseline, week 12, and follow-up (week 24):

1. The Treatment Self-Regulation Questionnaire (TSRQ) [39]. The TSRQ is designed to assess different forms of motivation that underlie an individual's behaviour, divided into autonomous and controlled domains. The responses were averaged according to each domain to reflect these motivational constructs individually (S2 Appendix).

2. The EQ-5D-5L Questionnaire (EuroQol) [40]. The EQ-5L-5D gives a subjective measure of participants' general health (S3 Appendix).

3. The Short Active Lives Survey [41]. The SALS measures self-reported levels of physical activity (S4 Appendix).

**Intervention group.**   The intervention group at the Westfield site, in addition to the above, had venous blood samples drawn at baseline (week 0), week 12, and week 24 to measure 10 biomarkers per blood draw. Participants received a written personalised and

motivational doctor's report relevant to the programme a week later detailing results of their test, accessible via an online portal with Medichecks (a UK-based direct-to-consumer blood testing and health check company). Apart from this, no incentive was offered to increase compliance of the intervention.

A subsample of participants was invited to complete an exit telephone interview to provide feedback on their experience of the study and whether any factors, such as blood testing, affected their motivation to change their lifestyle. These conversations were audio recorded and stored as a password-protected file on an encrypted laptop until transcribed verbatim.

**Blood collection and analysis.**   Non-fasting samples were collected from participants by certified phlebotomists on site. Venous samples (10ml) were collected from the antecubital fossa region using an aseptic technique into serum gel and EDTA tubes (Vacutainer®BD UK Ltd, Oxford, UK). Samples were stored at -80˚C, then received and processed by a United Kingdom Accreditation Service (UKAS) approved laboratory within 24 hours of blood draw.

Ten routine blood biomarkers were analysed: lipids (total cholesterol, low-density lipoprotein (LDL) cholesterol, high-density lipoprotein (HDL) cholesterol, non-HDL cholesterol, and triglycerides), HbA1c, high-sensitivity C-reactive protein (hs-CRP), vitamin D, active vitamin B12, and ferritin.

**Biomarker feedback.**   Biomarker results were accessible to participants within one week of blood draw via Medichecks' secure online portal. Baseline blood test results were also printed and distributed to participants at the beginning of their session in the second week. Results were displayed on a scale to clearly demonstrate whether each biomarker was low, normal, or raised. A doctor's commentary was provided with motivational insights on how to improve their results. If any result was grossly abnormal and required further investigation, these participants were notified and advised to follow up the result with their GP or to seek urgent medical advice if required. Any health issues precluding participants from continuing with the programme were excluded from the study.

## Primary objectives

**Feasibility outcomes (primary objectives 1–2).**   Feasibility of recruitment and retention were measured by the number of participants recruited to the study; the number of weekly sessions and blood testing sessions attended; questionnaire completion rates; and retention/attrition rates. Retention rate was defined as the percentage of recruited participants who did not withdraw from the program at 12 weeks, regardless of how many endpoint outcome measures were collected.

The percentage of participants from whom valid primary outcome measures were collected (including BMI, questionnaire results, and blood test results) was used to assess the feasibility of obtaining these measures. Blood collection was overseen by two research staff members, including a medical professional. Adverse events were recorded and any changes to the intervention were documented.

**Acceptability outcome (primary objective 3).**   Acceptability of the intervention was assessed using post-intervention interviews delivered by one of the research staff who was not involved in the intervention delivery. Questions were based on the theoretical framework of acceptability provided by Sekhon et al. [42] to include self-reported satisfaction and experience of the intervention and its effects on motivation and willingness to continue the programme. Thematic analysis was conducted to group responses.

**Sample size estimation (primary objective 4).**   Effect size (Cohen's $d$) was calculated as 0.344 by dividing the mean difference in BMI between the two groups (at baseline and 12 weeks) by the pooled standard deviation of the entire sample, which corresponded to a small-to-medium effect size.

### Secondary objectives

**Behavioural and clinical outcomes (secondary objectives 1 and 2).**   To assess whether biomarker feedback may have impacted participant behaviour within the programme, several outcomes were compared between the control and intervention groups, including session attendance; retention rates; and responses from the corrected TSRQ, Short Active Lives Survey, and exit interviews. BMI scores were also compared at baseline and 12 weeks.

Changes in biomarker data over the 12-week programme were assessed to establish whether biomarker feedback may provide useful information in addition to weight loss outcomes within the programme.

### Data collection and statistical analysis

**Baseline assessment and physical measurements.**   Consenting, eligible participants in both the intervention and control groups completed a baseline questionnaire administered by one of the research team. The questionnaire collected information on demographic characteristics (age, gender, and ethnicity). Baseline height and BMI, as well as subsequent measures, were recorded by Shape Up staff.

**Questionnaire data.**   Questionnaires were distributed by research staff and completed by participants in person at the first, 12-week, and follow-up sessions.

**Statistical analysis.**   Descriptive statistics with means, standard deviations (SD) and percentages were used to summarise the demographic characteristics of participants, survey scores, and biomarker results. The demographics of the control and biomarker feedback groups were compared using the independent samples two-tailed t-test for continuous descriptors with normal distributions and the two-tailed Fisher's exact test for ethnicity variables. For survey data, the mean difference between groups in change from baseline to 12 weeks was calculated on paired data. The independent two-tailed t-test was used to generate 95% confidence intervals. For changes in biomarker data between 0 and 12 weeks, mean difference and 95% confidence intervals were calculated using the paired two-tailed t-test. Given that this was an exploratory pilot study, p values were not provided, except to demonstrate comparability of baseline characteristics between groups (Table 1). While our primary focus was not on intervention effectiveness, we have reported preliminary outcome data for completeness and to inform future research. For these outcomes, the analysis was conducted on a modified intent-to-treat (ITT) basis, including all participants who were initially enrolled and randomised, and for whom complete baseline and follow-up data were available. All statistical analyses were carried out using R (R Core Team, 2023) [43].

Attendance was assessed as a total percentage for each participant over the 12-week programme and averaged per group.

### Deviations from the study protocol

In the original study protocol, participants were to be randomised between the control and biomarker feedback groups across both sites. However, due to logistical constraints and the impracticalities of implementing the intervention across both sites, each site formed the control group and intervention group, respectively. Participants continued with their programme which was closest to their place of residence, regardless of whether they joined the study. This arrangement should be taken into consideration when interpreting the study's findings, particularly the feasibility of introducing the intervention across two sites and recognising the potential influence of site-specific factors on the outcomes. Secondly, the Mental Component Summary of the 12- item Short Form (SF12), which measures emotional quality of life, was included in the original study protocol. This was removed as mental and

**Table 1. Baseline demographic characteristics of study participants.** The independent two-tailed t-test was used to calculate p values comparing continuous variables (age, weight, and BMI). The two-tailed Fisher's exact test was used to calculate p values comparing categorical data (ethnicity). SD: standard deviation; BMI: body mass index.

| | Control group (n = 24) | Biomarker feedback group (n = 22) | P value |
|---|---|---|---|
| **Age (years), mean±SD; range** | 45.4±7.4; 29–57 | 48.5±9.8; 30–65 | 0.2287 |
| **Weight (kg), mean±SD** | 108.5±19.2 | 112.8±22.1 | 0.5236 |
| **BMI (kg/m$^2$), mean±SD** | 33.7±4.0 | 35.1±4.3 | 0.2636 |
| **Ethnicity, n (%)** | | | 0.3400 |
| **White** | | | |
| *English, Welsh, Scottish, Northern Irish, or British* | 17 (70.8) | 16 (72.7) | |
| *Irish* | 2 (8.3) | 1 (4.5) | |
| *Other white background* | 0 (0) | 3 (13.6) | |
| **Asian or Asian British** | | | |
| *Indian* | 1 (4.2) | 1 (4.5) | |
| *Chinese* | 1 (4.2) | 0 (0) | |
| **Other ethnic group** | | | |
| *Arab* | 1 (4.2) | 0 (0) | |
| *Other mixed/multiple ethnic group* | 0 (0) | 1 (4.5) | |
| **Unknown** | 2 (8.3) | 0 (0) | |

emotional health was not the focus of the study. The three chosen questionnaires aim to gauge overall health, motivation, and levels of activity.

## Results

### Study population

Baseline characteristics of the participants by group are shown in Table 1. The overall mean ±SD age was 46.8±8.7 years and BMI 34.4±4.1 kg/m$^2$.

### Recruitment and retention (primary objective 1)

**Recruitment.** At the control group site, 26 (96.3%) out of all eligible 27 Shape Up participants were recruited, with one participant non-consenting (Fig 1). During the 12-week programme, 2 participants (2/26, 7.6%) withdrew from the Shape Up programme and were excluded from data analysis. Hence, 24 participants were considered for data analysis. At 24 weeks, 19 participants (19/24, 79.2%) were lost to follow-up, and no control participants attended their 24-week weigh-in. At the biomarker feedback site, 29 (85.3%) out of 34 participants declared their consent and were recruited. For data analysis, 7 participants (7/29, 24.1%) were excluded due to incompletion of the 12-week programme (4/29, 13.8%) and failure to attend at least the first blood test (3/29, 10.3%). Hence, 22 participants were considered for data analysis. At 24 weeks, 10 participants were lost to follow-up. There was no significant difference between retained participants' demographic characteristics and those that dropped out or were lost to follow-up (p>0.05 for all variables).

**Retention and attendance.** Mean retention and attendance rates over the course of the 12-week programme were 92.3% and 83.7% in the control group, and 86.2% and 81.5% in the biomarker feedback group, respectively. However, 3 of these retained participants in the biomarker feedback group were excluded from analysis due to no blood test results recorded at baseline and other incomplete outcome measures.

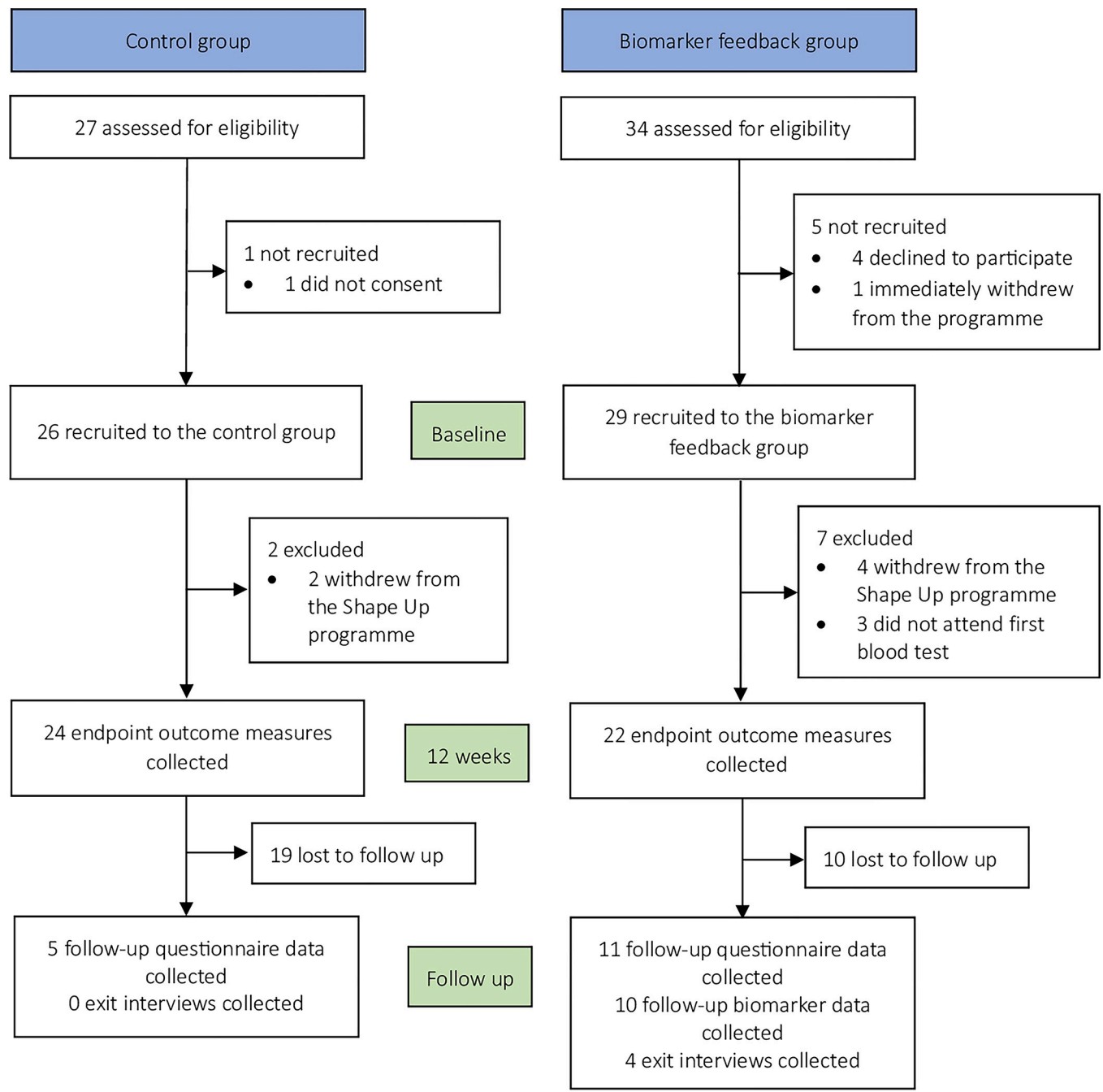

**Fig 1. Trial profile showing number of participants screened, excluded, recruited, and analysed.**

## Collection of behavioural and clinical outcome measures (primary objective 2)

**Questionnaires and exit interviews.** Questionnaire completion rates at baseline, 12 weeks, and follow up were 75.0%, 54.2%, 20.8% in the control group, and 90.9%, 86.3%, 50.0% in the biomarker feedback group, respectively. Exit interviews were completed by 4 participants in the intervention group.

**Blood collection.**    Of the 22 participants in the biomarker feedback group, 19 (86.4%) attended their second blood test at 12 weeks, and 10 (45.5%) attended their follow-up blood test at 24 weeks. Due to raised triglyceride results, 4 LDL cholesterol results (1 at baseline, 3 at 12 weeks) could not be calculated. One participant's 12-week hs-CRP result and all 3 ferritin results were excluded from analysis due to illness and a subsequent diagnosis of haemochromatosis.

All participants that attended the blood collection sessions received their results via Medichecks' online portal within one week alongside a doctor's comments. No adverse outcomes were recorded during the blood taking sessions on site.

## Participant acceptability outcomes (primary objective 3)

In general, participants were positive towards blood collection and biomarker feedback. Their exit interview responses were grouped according to different aspects of the intervention.

**Blood testing and logistics.**    Most participants perceived the intervention to be an acceptable addition to the programme.

*"No bother. Nice, straightforward—didn't take too long. I think the first night was probably a bit time-consuming in terms of logistics, but the girls and guys doing it were very friendly, very nice."* [Participant 1]

*"I thought it was a really good idea to. . .help me and help others."* [Participant 2]

*"For some reason I don't give up my blood very easily, but that didn't stop me being prepared to do it anyway. Having information is an important thing."* [Participant 3]

*"Fine. No issues."* [Participant 4]

**Results delivery.**    There were mixed views on the delivery of results. Some participants preferred receiving their results online rather than in the session.

*"If there's a way to get the results faster, that's better, and if you get them privately and individually, then you have time to digest them. It won't take time out of the programme."* [Participant 1]

*"We had elements of competition between us comparing the blood tests."* [Participant 2]

*"When you hand them out you can get a bit upset when your result is bad, but others are good because everyone compares them. [. . .] I didn't enjoy getting the printout in the group because my results were bad."* [Participant 3]

*"I think that if the results are delivered in the right way, then it can be motivating and not scary, even if it's bad. [. . .] Having a medic or scientist to deliver the [group] feedback would have been useful."* [Participant 4]

**Doctor's report and results.**    One participant commented on how abnormal results may come as a shock, and that a phone call may be helpful for reassurance.

*"When the results came, the results were a bit matter of fact and there were some quite serious implications from them. I've just had this bombshell and I'm a little bit shell-shocked. I think if you get some important results back then a call would be useful and reassuring."* [Participant 3]

However, they and other participants added that the doctor's report was beneficial.

*"The motivational report was really useful—you can do this or that. Don't panic, you can do this. Much better."* [Participant 3]

*"They were pretty concise. Gave an explanation of ranges. Graphs and colour would be useful going forward to get the message home. The first one definitely was a big incentive to do something about it."* [Participant 1]

*"They were [clear and understandable], but then I have a scientific mind so I'm always looking at things like this."* [Participant 2]

## Estimating sample sizes needed to power a future RCT (primary objective 4)

Assuming a power of 80%, significance level of 0.05, and two-tailed hypothesis, a minimum of 320 participants (160 participants per group) would need to be recruited to adequately power an RCT, accounting for an expected dropout rate of 20%.

## The effect of biomarker feedback on clinical outcomes and behavioural outcomes (secondary objective 1)

**Weight and BMI.** Approximately 68.1% of the biomarker feedback group and 58.3% of the control group lost ≥5% of their initial body weight.

Mean±SD reduction in BMI scores over the 12-week programme was greater in the biomarker feedback group (2.1±1.2; 6.0%) than in the control group (1.7±1.2; 5.0%) (Fig 2). High attrition rates resulted in insufficient data for analysis at follow-up.

**Behavioural outcome measures.** SALS scores represent self-reported measures of the number of minutes of moderate-intensity physical activity carried out per week, defined as activities that increase breathing rate. Mean SALS scores improved in both groups over the course of the programme, most notably in the biomarker feedback group (Table 2). For paired data, this represented a 70.1% increase in the control group, and 128.9% increase in the biomarker feedback group.

Mean autonomous and controlled TSRQ scores increased minimally in the control group. In the biomarker feedback group, mean controlled TSRQ scores (27.6%) increased more so than mean autonomous scores (7.6%) from baseline to programme end for participants with paired data. By follow-up, mean autonomous TSRQ scores in the biomarker feedback group had decreased for the 11 participants with paired data (MD 0.5 [95% CI –0.20, 1.20]).

The percentage increase in mean EQ-5D-5L scores was greater among biomarker feedback group participants (60.7%) than control group participants (34.9%) for participants with paired data at baseline and programme end. Mean EQ-5D-5L scores at follow-up were similar to programme-end scores in both groups.

**Exit interview data.** All 4 participants from the biomarker feedback group who completed an exit interview shared a view that biomarker feedback could be motivating when delivered in the right way. Seeing an improvement in biomarker results was noted to be more effective than the results taken in isolation for most participants.

*"The first one was a big incentive to do something about it. The second one you get the thought of 'I don't want to stop this now' and you want to carry on."* [Participant 1]

*"It was really encouraging. When we compared the first set of tests which were very similar to the bloods I had just done with the GP, and then compared them with the test at 3 months there was massive improvement across the board."* [Participant 2]

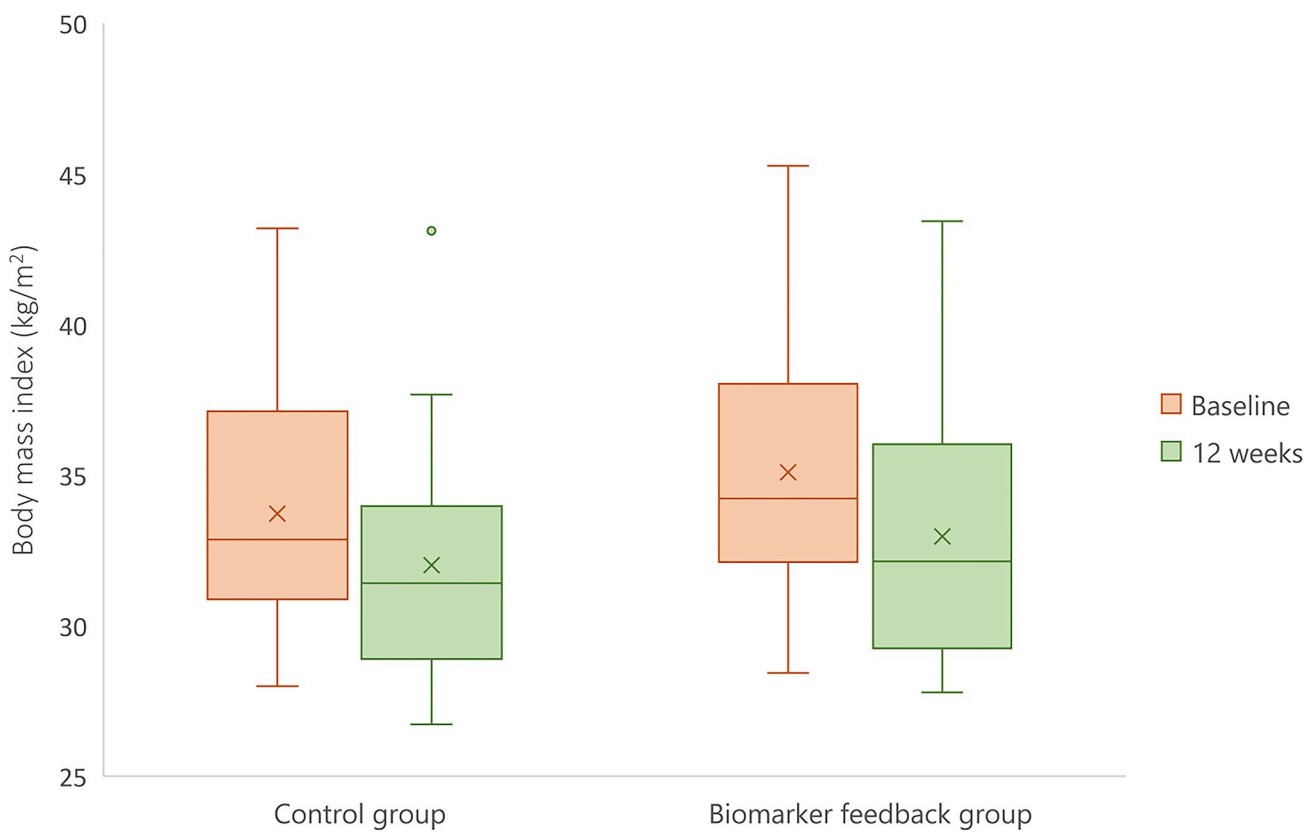

**Fig 2. Box plot to show changes in BMI across the control group (n = 24) and biomarker feedback group (n = 22) at baseline and at the end of the 12-week programme.** The boxes bound the interquartile range (IQR) divided by the median, and whiskers extend to a maximum of 1.5 × IQR beyond the box. Crosses represent the mean.

*"Knowing that it was working with follow-up tests was the motivation. It was satisfying and a relief. And it was good to see how quickly it can happen." [Participant 3]*

*"I think that if the results are delivered in the right way, then it can be motivating and not scary, even if it's bad. [. . .] I think things like the blood tests helps because you kinda [sic] want to see there's an improvement. As much as the weigh-ins help, it's the blood tests which give you reaffirmation that it is working and extra motivation for the mid-way or end goal." [Participant 4]*

**Table 2. Survey scores of the study groups at baseline and programme end (week 12).** Negative values indicate that improvements in survey scores exceeded those in the control group; *b* = mean difference between groups in change from baseline to 12 weeks for paired data; 95% CI = confidence intervals of the mean difference calculated using the independent two-tailed t-test; and *d* = *Cohen's d*. SD = standard deviation; SALS = Short Active Lives Survey; TSRQ = Treatment Self-Regulation Questionnaire; EQ-5D-5L = EuroQol-5D-5L.

| | Control group | | Intervention | | *b* | 95% CI | *d* |
|---|---|---|---|---|---|---|---|
| | **Baseline** | **Week 12** | **Baseline** | **Week 12** | | | |
| **SALS, mean±SD; n** | 190.56±242.50; 18 | 335.00±211.01; 13 | 359.55±362.44; 20 | 830.00±585.50; 19 | -329.24 | -606.91, -51.57 | -0.82 |
| **TSRQ autonomous, mean±SD; n** | 5.27±1.57; 18 | 5.83±1.43; 13 | 5.88±0.94; 20 | 6.35±0.57; 19 | -0.20 | -1.07, -0.67 | -0.17 |
| **TSRQ controlled, mean±SD; n** | 2.53±1.15; 18 | 2.81±1.49; 13 | 2.91±1.38; 20 | 3.70±1.14; 19 | -0.51 | -1.43, -0.40 | -0.41 |
| **EQ-5D-5L, mean±SD; n** | 59.94±18.89; 18 | 75.75±8.92; 12 | 47.75±19.97; 20 | 71.94±12.85; 18 | -7.64 | -17.89, -2.61 | -0.57 |

**Table 3. Biomarker results at baseline, programme end, and follow up in the biomarker feedback group.** All figures are given as mean ± standard deviation unless otherwise stated. Negative values represent an increase between baseline and 12-week results. The paired two-tailed t-test was used to calculate 95% confidence intervals. P values were omitted with the primary focus of this study being feasibility; $d$ = *Cohen's d*. *Includes only participants for whom paired baseline and 12-week biomarker data were collected. **3 results at baseline and 1 result at 12 weeks excluded from analysis as triglycerides were too high to calculate an LDL cholesterol result. †1 participant was unwell, so anomalous hs-CRP result was excluded from analysis. ‡1 participant was found to have haemochromatosis so was excluded from analysis. LDL = low-density lipoprotein; HDL = high-density lipoprotein; TC = total cholesterol; hs-CRP = high-sensitivity C-reactive protein.

| | Biomarker feedback group | | | | |
| --- | --- | --- | --- | --- | --- |
| | **Baseline** | **Programme end(week 12)** | **Follow up (week 24)** | **Mean difference (baseline vs programme end*) (95% CI)** | $d$ |
| **HbA1c (mmol/mol); n** | 39.20±6.75; 22 | 35.98±4.09; 19 | 36.40±2.68; 11 | 3.89 (1.54, 6.23) | 0.58 |
| **Total cholesterol (mmol/L); n** | 5.26±0.84; 22 | 4.95±1.05; 19 | 4.73±1.03; 11 | 0.43 (0.06, 0.81) | 0.44 |
| **Non-HDL cholesterol (mmol/L); n** | 4.14±0.80; 22 | 3.73±1.00; 19 | 3.52±0.94; 11 | 0.52 (0.15, 0.88) | 0.56 |
| **LDL cholesterol (mmol/L); n** | 2.98±0.71; 19** | 2.91±0.85; 18** | 2.51±0.85; 11 | 0.13 (-0.21, 0.47) | 0.15 |
| **HDL cholesterol (mmol/L); n** | 1.11±0.29; 22 | 1.22±0.29; 19 | 1.21±0.38; 11 | -0.08 (-0.01, -0.15) | -0.28 |
| **TC:HDL cholesterol ratio; n** | 4.98±1.51; 22 | 4.24±1.24; 19 | 4.15±1.26; 11 | 0.80 (0.42, 1.18) | 0.51 |
| **Triglycerides (mmol/L); n** | 2.85±1.73; 22 | 1.94±1.15; 19 | 2.23±1.05; 11 | 1.07 (0.60, 1.54) | 0.57 |
| **QRISK2 score (%); n** | 7.27±5.52; 20 | 5.30±3.30; 19 | 5.48±5.71; 11 | 1.30 (-0.07, 2.66) | 0.28 |
| **hs-CRP (mg/L); n** | 4.80±4.54; 22 | 3.92±5.87; 18† | 3.25±3.08; 10 | 0.99 (-0.60, 2.59) | 0.18 |
| **Vitamin D (nmol/L); n** | 41.65±15.82; 22 | 59.04±17.75; 19 | 59.00±14.85; 10 | -15.38 (-9.94, -20.83) | -0.91 |
| **Active vitamin B12 (pmol/L); n** | 94.85±44.93; 22 | 80.30±31.21; 19 | 78.34±15.16; 10 | 7.71 (-0.44, 15.86) | 0.25 |
| **Ferritin (µg/L); n** | 343.00±248.09; 21‡ | 294.76±203.42; 18‡ | 293.18±188.18; 9‡ | 65.46 (22.76, 108.16) | 0.21 |

One participant commented on how group biomarker feedback helped to boost group morale.

*"I think what was good was the collective–we all as a whole group were borderline diabetic and cholesterol levels, so it was a group thing to move forwards together. You could really see it was because of fitness and diet. And then later with the next results you could see the pattern of improvement with everyone, which confirmed what we were doing was working." [Participant 4]*

### Biomarker results (secondary objective 2)

The within-group mean difference between baseline and programme end suggests that the physical activity programme may have favourable effects on HbA1c, total cholesterol, non-HDL cholesterol, HDL cholesterol, TC:HDL cholesterol ratio, triglycerides, and vitamin D levels (Table 3).

### Discussion

This pilot study demonstrates that it is feasible to recruit and retain participants in a 12-week physical activity programme with a biomarker feedback intervention, which is acceptable to participants. While this study was not sufficiently powered, the findings suggest potential favourable effects of biomarker feedback on weekly physical activity levels and certain aspects of motivation. Additionally, using biomarker data as an outcome measure may be useful to assess the impact of a physical activity programme beyond weight measurements alone.

Recruitment and retention rates for the control group were high at 96.3% and 92.3%, respectively. This was noticeably higher than the biomarker feedback group whose recruitment and retention rates were 85.3% and 86.2%. It is possible that the additional time

burden of a blood test or needle phobia, which may be as high as 20–30% in adults [44], discouraged some participants from taking part in the study. However, of the 22 participants analysed within the biomarker feedback group, 19 (86.4%) attended their second blood test. Beyond the 12-week Shape Up programme, it became increasingly difficult to engage participants in data collection. Retention rates at follow-up (24 weeks) were 20.8% and 50.0% in the control and biomarker feedback group, respectively. Therefore, the longer-term effects of biomarker feedback in these types of community programmes may be more challenging to assess.

Questionnaire completion rates were lower than expected in the control group (75.0% at baseline, 54.2% at 12 weeks) compared with the biomarker feedback group (90.9% at baseline, 86.4% at 12 weeks). Receiving biomarker feedback may have incentivised participants to better engage with the study and questionnaires. Compensatory strategies for low response rates in future trials might include collecting the data on site rather than an online form, sending reminder emails to participants, or providing an incentive on completion.

Overall, participants considered blood collection and biomarker feedback to be an acceptable intervention. All 4 participants who were interviewed provided positive feedback, stating that biomarker feedback was both informative, useful, and motivational. Participants generally preferred to receive their results privately, rather than in a group setting to avoid potential feelings of comparison or judgment. However, giving feedback on the group's average results may be helpful to boost group morale. Doctors' comments were beneficial, motivational, and generally reassuring. One participant, whose HbA1c level was significantly raised, reported their results came as a shock and would have preferred to receive their results over a phone call. While this approach would offer participants an opportunity to ask questions about results that are significantly out of range or warrant further investigation, the resources needed for this should be considered, especially if explored on a larger scale.

This study suggests that biomarker feedback has mixed effects on participant behaviour and motivation within the Shape Up programme. While changes in BMI were similar between the two groups, self-reported measures of physical activity were markedly increased in the biomarker feedback group compared to the control group. Interview data suggests that one possible reason for this is that biomarker feedback may act as an incentive to continue healthy behaviours. Mean TSRQ scores increased in both groups. In the biomarker feedback group, controlled scores increased to a greater extent than autonomous scores, which has been associated with non-adherence to treatment and poorer well-being [45].

While measures of adherence, such as retention and attendance, were lower among the biomarker feedback group, improvements in self-reported levels of activity (SALS) were greater than the control group, as well as general measures of wellbeing (EQ-5D-5L). The significance of these findings cannot be determined with small sample sizes and warrants further exploration. Some participants reported that the most motivational aspect of biomarker feedback came from comparing two sets of results that provided evidence of improvement. Unfortunately, low retention rates at follow-up makes it difficult to explore the cumulative effect of biomarker data on behaviour and motivation. It is also worth considering whether this may introduce potential drawbacks of incorporating biomarker feedback into a physical activity programme. Just as improvements in blood test results may improve motivation, blood test results could adversely affect health behaviour, especially if little or no improvement is seen, or if results are already within a healthy range. For example, if cholesterol results are normal, it may discourage individuals from shifting to a healthier diet [46]. Such an effect is unlikely in this study, since most participants' results improved, but this should be a consideration if it is investigated in other settings. Further, one should note that the effect of biomarker feedback on a participant's levels of motivation is likely to be highly

individual. Unlike weight or BMI, which often trends downwards, a panel of blood test results focuses on multiple aspects of health, from diabetes risk to cholesterol levels. Some areas of health may be more important to some participants than others and this should be investigated in future studies.

Aside from its acceptability and effects on motivation, participants also reported on additional benefits that biomarker feedback brought to the programme. Firstly, it provided health insights that would otherwise have gone unrecognised: *"The hidden benefits can't be seen unless you do this sort of thing."* [Participant 4] and *"If I hadn't had that, I perhaps wouldn't have had the scientific proof that it's working."* [Participant 2]. For some individuals, their blood test results prompted further investigations which led commencement of treatment or a new diagnosis: *"I'm on statins and if I hadn't decided to get checked out, it wouldn't have happened."* [Participant 2] and *"For me, I have haemochromatosis and I wouldn't have found out about that."* [Participant 4]. One individual was diagnosed with diabetes at the beginning of the programme and was able to reduce their HbA1c to a normal level by the end of the programme. Secondly, biomarker feedback before and after the Shape Up programme provided further reassurance that the programme was effective: *"The second one you get the thought of 'I don't want to stop this now' and you want to carry on."* [Participant 1] and *"As much as the weigh-ins help, it's the blood tests which give you the reaffirmation that it is working and extra motivation for the mid-way or end goal."* [Participant 4]. Thirdly, it is possible that biomarker data may help to tailor aspects of the programme to the individual: *"Not in the exercise part but in the diet part of the programme, he was able to relate back to the tests."* [Participant 4].

Marked improvements were seen in several biomarkers, including HbA1c, triglycerides, and vitamin D. These data help to highlight the effectiveness of the programme, especially in participants with only slight reductions in their BMI over the 12-week programme. For example, one participant's BMI improved marginally, from 28.4 to 27.8 kg/m$^2$, whereas there was a more marked improvement in his HbA1c (-2.4mmol/mol) and triglyceride (-1.36 mmol/L) levels. Another participant's HbA1c result reduced from 65.52 mmol/mol (baseline) to 43.73 mmol/mol (12 weeks) which could help illustrate the cost-saving potential of the programme for conditions like diabetes. For project organisers, biomarker feedback may help validate their programme, offering an opportunity to tailor parts of the programme to the individual and to strengthen funding applications by adding to existing evidence of the programme's efficacy.

Strengths of this study include blood collection at three timepoints including follow-up, with 10 biomarkers analysed. This study is also one of few investigating the effect of biomarker feedback on participation in a physical activity programme, which also contains a control group. There were several limitations to this study. Participants were recruited and assigned to a group based on location. Such clustering leaves data open to confounding factors that might influence engagement in the programme, such as group morale, inter-individual competition, or small differences in the way the sessions were delivered. Participants' baseline levels of motivation were not captured. It is uncertain how a randomised study design with biomarker feedback across all sites might affect recruitment, retention, and the collection of valid outcome measures. Furthermore, the focused demographics of participants and limited ethnicity data restricts generalisability of feasibility across more diverse groups. For example, some evidence shows that levels of adherence are lower among women [12]. While biomarker feedback was generally perceived as an acceptable intervention, only a small percentage of participants completed an exit interview. For those participants who withdrew from the study, interview data was not captured, which may have provided retention insights for future studies. The absence of follow-up data for some participants required a focus on complete

cases. This modified ITT approach may introduce some bias due to attrition and should be carefully considered in the design of future, larger-scale studies where strategies for data imputation may be employed to address missing data in accordance with strict ITT analysis. Most measures of motivation included in this study were self-reported, such as questionnaires and interviews, which may be subject to bias. In addition to BMI, attendance, and retention, other objective measures may help to strengthen data, such as physical activity trackers.

## Conclusion

This pilot study indicates the feasibility of recruiting and retaining men in a physical activity programme with questionnaire data as an outcome measure and biomarker feedback as an acceptable intervention. Though this study is not sufficiently powered, preliminary data suggests that biomarker feedback may improve physical activity levels and some aspects of motivation within a physical activity programme. There is limited evidence to suggest that biomarker feedback improves attendance or retention rates within the programme. Improvements in biomarker results may provide additional evidence of programme efficacy in addition to weight loss. Further larger-scale trials are required to assess the significance of these findings, subject to identified refinements.

## Supporting information

**S1 Appendix. TRENDS checklist.**
(PDF)

**S2 Appendix. Treatment questionnaire concerning continued program participation.**
(DOCX)

**S3 Appendix. EQ.5D.**
(PDF)

**S4 Appendix. Short Active Lives Survey.**
(PDF)

## Acknowledgments

We thank the participants, Watford FC, and the study staff of the Shape Up programme. We also acknowledge Samha Al-Mazi, medical student at UCL, for her contributions to the data analysis.

## Author Contributions

**Conceptualization:** Daniel Grant, Lindsay Bottoms.

**Data curation:** Daniel Grant, Joshua Smith, Lindsay Bottoms.

**Formal analysis:** Joshua Smith.

**Investigation:** Daniel Grant, Lindsay Bottoms.

**Methodology:** Daniel Grant, Joshua Smith, Lindsay Bottoms.

**Project administration:** Joshua Smith, Lindsay Bottoms.

**Resources:** Daniel Grant, Lindsay Bottoms.

**Writing – original draft:** Joshua Smith.

**Writing – review & editing:** Daniel Grant, Joshua Smith, Lindsay Bottoms.

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
