## [Decision Letter · Decision Letter 0]

23 Jan 2024

PONE-D-23-34787Assessing the role of biomarker feedback in a 12-week community weight management programme among overweight men: a pilot studyPLOS ONE

Dear Dr. Bottoms,

Thank you for submitting your manuscript to PLOS ONE. After careful consideration, we feel that it has merit but does not fully meet PLOS ONE’s publication criteria as it currently stands. Therefore, we invite you to submit a revised version of the manuscript that addresses the points raised during the review process.

We look forward to receiving your revised manuscript.

Kind regards,

David Chibuike Ikwuka, Ph.D.

Academic Editor

PLOS ONE

Journal Requirements:

Reviewers' comments:

Reviewer's Responses to Questions

**Comments to the Author**

1. Is the manuscript technically sound, and do the data support the conclusions?

Reviewer #1: Partly

Reviewer #2: Yes

2. Has the statistical analysis been performed appropriately and rigorously? 

Reviewer #1: No

Reviewer #2: Yes

3. Have the authors made all data underlying the findings in their manuscript fully available?

Reviewer #1: Yes

Reviewer #2: Yes

4. Is the manuscript presented in an intelligible fashion and written in standard English?

Reviewer #1: Yes

Reviewer #2: Yes

5. Review Comments to the Author

Reviewer #1: Line 28, Line 357, Line 459, the abbreviation SD to be placed after the word mean.

Line 214, the registration number to be stated.

Line 229, the follow up period to be stated e.g. week 24

It is not clear how the subjects after fulfilling the inclusion and exclusion are assigned or grouped as intervention and control though it was a non-randomization study. More description is to be provided.

A description of the control group is to be provided apart from describing the intervention group. What is similar/different in terms of intervention and assessment.

Figure 1, the sites Westfield and Borehamwood which formed the intervention and control groups respectively are to be stated on top first to begin with after which the subjects were assigned to control and intervention groups.

Line 293, the statement on sample size calculation was not required (Line 206-208) and the statement ‘estimating sample sizes needed to power a future RCT’ (Line 294-296, Line 444) requires connection flow of presentation.

Line 324 n (%) was also presented other than mean(sd).

Line 325-327, the sentence requires revision.

Line 326, the exact type of t-test is to be stated.

Line 328, one or two-tailed test for Fisher’s Exact Test is to be stated.

The use of Mann-Whitney U test to be clearly denoted in the results section.

A footnote to denote those empty p values is to be provided.

Analyses whether intent to treat or per protocol analyses are to be stated in the statistical analyses section.

At least 1 decimal point for the percentage figures to be presented in the text and tables.

Line 331-332, although this statement 'Given that this was an exploratory pilot study, p values were not provided.' was referring to Table 2 and 3, the statement is contrast with Table 1 where p values were given.

Table 1,2, 3 statistical test(s) to be denoted in the table footnote.

Table 3 hs-CRP (mg/L), two decimal points for 4.8 is to be provided

Effect size index could be presented.

Some references did not conform to the journal format.

Reviewer #2: Line 68 - Were weight management programs available prior to the pandemic/2020? If so, did enrollment and completion rates change pre-/post-pandemic?

Line 332 - R software needs to be cited, not R studio

R Core Team (2022). R: A language and environment for statistical computing. R Foundation for Statistical Computing, Vienna, Austria. URL https://www.R-project.org/.

Line 483/table 2 - The table is difficult to read due to the lack of spacing and inconsistencies in cell formatting.

Line 486/table 2 - BMI is listed as one of the measures but it is not in the table

6. PLOS authors have the option to publish the peer review history of their article (what does this mean?). If published, this will include your full peer review and any attached files.

Reviewer #1: No

Reviewer #2: No

---

## [Author Response · Author response to Decision Letter 0]

1 Feb 2024

Comments from reviewers

We would like to thank the reviewers for taking the time to go through our manuscript and provide feedback and suggestions. We have gone through and responded to each point and amended the manuscript. We believe the paper is improved and thank the reviewers for their useful input. 

Reviewer #1: Line 28, Line 357, Line 459, the abbreviation SD to be placed after the word mean.

Abbreviation SD added after mention the mean on line 28, 357 (377 in tracked version) and 459 (483 in tracked version). 

Line 214, the registration number to be stated.

Registration number added in brackets on lines 217-218 in tracked version. 

Line 229, the follow up period to be stated e.g. week 24

Week 24 added in brackets to line 235 in tracked version.

It is not clear how the subjects after fulfilling the inclusion and exclusion are assigned or grouped as intervention and control though it was a non-randomization study. More description is to be provided.

Each site was randomly chosen to be either the control or intervention group. A line has been added (205-206 tracked version) to reflect this. This limitation of the study methods is mentioned in the discussion.

A description of the control group is to be provided apart from describing the intervention group. What is similar/different in terms of intervention and assessment.

The ”Intervention” section (line 220 in the tracked version) has now been divided into a control group and an intervention section (lines 246 in tracked version) to make these differences clearer. 

Figure 1, the sites Westfield and Borehamwood which formed the intervention and control groups respectively are to be stated on top first to begin with after which the subjects were assigned to control and intervention groups.

Figure updated to show that each site was assigned to be the control and intervention group in addition to the description in lines 205-206. This figure has also been updated to clarify the number of endpoint measures which were not provided in the first draft. 

Line 293, the statement on sample size calculation was not required (Line 206-208) and the statement ‘estimating sample sizes needed to power a future RCT’ (Line 294-296, Line 444) requires connection flow of presentation.

Line 293 (line 305-306 in tracked version) has been deleted. 

We’ve also deleted repeated material in the results section (lines 471-473 in tracked version) so that these sections flow better. 

Line 324 n (%) was also presented other than mean(sd).

This sentence now includes “and percentages” (line 334 in tracked version). 

Line 325-327, the sentence requires revision.

Lines 337-353 (in the tracked version) have been updated, including this sentence which has been broken down for clarity. 

Line 326, the exact type of t-test is to be stated.

Clarification added that this is an independent samples two-tailed t-test (lines 336-337 in tracked version).

Line 328, one or two-tailed test for Fisher’s Exact Test is to be stated.

“Two-tailed” added to lines 337-338 (tracked version)

The use of Mann-Whitney U test to be clearly denoted in the results section.

In order to provide mean difference and confidence intervals, we have used the paired two-tailed t-test for biomarker results between baseline and 12 weeks. This is reflected in the updated copy in line 340-342 (tracked version). 

A footnote to denote those empty p values is to be provided.

Added in the footnote of the blood test results table why p values have not been included, namely because this is a feasibility study (lines 555-556 in tracked version). 

Analyses whether intent to treat or per protocol analyses are to be stated in the statistical analyses section.

Two sentences to demonstrate this was a modified ITT approach (lines 348-352 in tracked version), and a further description of the limitations of this has been added to the discussion (lines 673-676 in tracked version).

At least 1 decimal point for the percentage figures to be presented in the text and tables.

Percentage figures updates throughout the text to include 1 decimal place. Results in table have been updated to 2 decimal places. 

Line 331-332, although this statement 'Given that this was an exploratory pilot study, p values were not provided.' was referring to Table 2 and 3, the statement is contrast with Table 1 where p values were given.

Additional sentence added: “except to demonstrate comparability of baseline characteristics between groups (Table 1) (lines 347-348 in tracked version). 

Table 1,2, 3 statistical test(s) to be denoted in the table footnote.

Table 1 footnote now mentions that this is an independent two-tailed t-test for continuous variables and Fisher’s exact test for categorical data. Table 2 footnote now mentions an independent two-tailed t-test is used to assess the mean difference between groups in change from baseline to 12 weeks for paired data. Table 3 footnote now mentions that the paired two-tailed t-test was used to generate confidence intervals comparing biomarker results at 0 and 12 weeks. 

Table 3 hs-CRP (mg/L), two decimal points for 4.8 is to be provided

2 decimal places now provided for all figures in table 3. 

Effect size index could be presented.

Effect size (Cohen’s d) now included in table 2 and 3 and clarification of negative values included in the footnotes. 

Some references did not conform to the journal format.

Edited so that all references follow PLOS ONE reference guidance including recognised journal abbreviations, date and authorship formats. 

Reviewer #2: Line 68 - Were weight management programs available prior to the pandemic/2020? If so, did enrollment and completion rates change pre-/post-pandemic?

The purpose of the study was to assess feasibility of integrating biomarker feedback into weight management programmes. Though interesting, we feel this information is less relevant to our study. Figures quoted in the introduction include both pre and post-pandemic figures, both of which could be improved.

Since writing the report, the preliminary public health figures have been updated. We have updates these (and the reference) in lines 67-69 (tracked version), which should a an increase in enrolments rated from 58% to 65%, but a decrease in completion rates (38% to 35%) and 5% weight loss (17% to 15%). Therefore, there is still room for improvement in terms of enrolment, completion, and success of these programmes. 

Line 332 - R software needs to be cited, not R studio

R Core Team (2022). R: A language and environment for statistical computing. R Foundation for Statistical Computing, Vienna, Austria. URL https://www.R-project.org/.

R software has now been referenced according to their website (line 352 tracked version) and reference added in line 43 (tracked version). 

Line 483/table 2 - The table is difficult to read due to the lack of spacing and inconsistencies in cell formatting.

Table 2 has been updated to no include Cohen’s d and formatting of table has been updated to make it easier to read. 

Line 486/table 2 - BMI is listed as one of the measures but it is not in the table

BMI had been added in error and this has now been removed from Table 2’s footnote.

---

## [Editor Report · Decision Letter 1]

13 Feb 2024

Assessing the role of biomarker feedback in a 12-week community weight management programme among overweight men: a pilot study

PONE-D-23-34787R1

Dear Dr. Bottoms,

We’re pleased to inform you that your manuscript has been judged scientifically suitable for publication and will be formally accepted for publication once it meets all outstanding technical requirements.

Kind regards,

David Chibuike Ikwuka, Ph.D.

Academic Editor

PLOS ONE

Additional Editor Comments (optional):

Thank you for your prompt response to the reviewers' comments and for submitting the revised version of your manuscript titled "Assessing the role of biomarker feedback in a 12-week community weight management programme among overweight men: a pilot study." I have carefully reviewed the revised manuscript and the responses to the reviewers' comments.

I am pleased to inform you that your revisions have addressed the concerns raised by the reviewers adequately. The changes made to the manuscript have significantly improved its clarity, organization, and scientific rigor.
---

## [Editor Report · Acceptance letter]

19 Mar 2024

PONE-D-23-34787R1 

PLOS ONE

Dear Dr. Bottoms, 

I'm pleased to inform you that your manuscript has been deemed suitable for publication in PLOS ONE. Congratulations! Your manuscript is now being handed over to our production team.

Kind regards, 

on behalf of

Dr David Chibuike Ikwuka 

Academic Editor

PLOS ONE